# Localisation-Dependent Variations in Articular Cartilage ECM: Implications for Tissue Engineering and Cartilage Repair

**DOI:** 10.3390/ijms26199331

**Published:** 2025-09-24

**Authors:** Laura Weimer, Luisa M. Schmidt, Gerhard Sengle, Marcus Krüger, Alan M. Smith, Ilona Brändlin, Frank Zaucke

**Affiliations:** 1Faculty 2: Computer Science and Engineering, Frankfurt University of Applied Sciences, 60318 Frankfurt, Germany; laura.weimer@fra-uas.de (L.W.); ilona.braendlin@fra-uas.de (I.B.); 2Department of Pharmacy, School of Applied Sciences, University of Huddersfield, Huddersfield HD1 3DH, UK; a.m.smith@hud.ac.uk; 3Institute for Genetics, Cologne Excellence Cluster on Cellular Stress Responses in Ageing-Associated Diseases (CECAD), and Center for Molecular Medicine Cologne (CMMC), University of Cologne, 50931 Cologne, Germany; luisa.schmidt@cpr.ku.dk (L.M.S.); marcus.krueger@uni-koeln.de (M.K.); 4Novo Nordisk Foundation Center for Protein Research, Department of Cellular and Molecular Medicine, Faculty of Health and Medical Sciences, University of Copenhagen, 1172 Copenhagen, Denmark; 5Department of Pediatrics and Adolescent Medicine, Faculty of Medicine and University Hospital Cologne, Center for Biochemistry, Center for Molecular Medicine Cologne (CMMC), Cologne Center for Musculoskeletal Biomechanics (CCMB), and Cologne Excellence Cluster on Cellular Stress Responses in Ageing-Associated Diseases (CECAD), University of Cologne, 50931 Cologne, Germany; gsengle@uni-koeln.de; 6Dr. Rolf M. Schwiete Osteoarthritis Research Unit, Department of Trauma Surgery and Orthopedics, University Hospital, Goethe University, 60528 Frankfurt, Germany

**Keywords:** articular cartilage, extracellular matrix, collagen, proteoglycan, proteomics

## Abstract

Articular cartilage (AC) is a specialised connective tissue covering joint surfaces. It enables smooth movement, distributes mechanical loads, and protects the underlying bone. In response to loading, AC adapts by modifying both its thickness and composition. AC is organised in different zones, with low cellularity and a high abundance of extracellular matrix (ECM). Mechanical overloading or immobilisation can lead to structural changes, potentially resulting in osteoarthritis (OA), for which no causal treatment currently exists. However, smaller defects can be treated using chondrocyte/cartilage transplantation or tissue engineering. A better understanding of the molecular composition of AC at different locations is essential to improve such therapeutic approaches. For this purpose, we performed a comprehensive analysis of porcine femoral knee cartilage at eight defined anatomical sites. Cartilage thickness and proteoglycan (PG) content were analysed histologically, while specific ECM proteins were assessed by proteomics and validated by immunohistochemistry and Western blot. Significant differences were identified, particularly between medial and lateral compartments, in terms of cartilage thickness, PG abundance, and ECM composition. Some proteins also showed zone-specific localisation patterns. These structural differences likely reflect adaptation to mechanical loading and should be considered to optimise future cartilage repair and tissue engineering strategies.

## 1. Introduction

Articular cartilage (AC) plays a central role in the function of the musculoskeletal system, as it covers the joint surfaces and provides a lubricated surface which enables the frictionless movement of bones [1,2]. It is a specialised connective tissue that provides essential functions for maintaining joint health, particularly through its mechanical properties, which allow it to facilitate the transfer of loads [1,3].

The structural and functional properties of AC are strongly dependent on its unique composition and organisation. It is relatively acellular, with only about 1–5% chondrocytes, the only cell type in the tissue, and about 95% extracellular matrix (ECM) [1,2,4].

The ECM is primarily composed of collagen fibres and proteoglycans (PG) [1,5]. Collagen type II (COL II) is the dominant structural protein that forms fibres together with collagen types IX and XI. These fibres provide cartilage with tensile strength and stability [1]. PGs, particularly aggrecan, are the main components of the amorphous gel-like ground substance, contributing significantly to the compressive tissue viscoelasticity. They achieve this by binding water through their negatively charged glycosaminoglycan chains, thereby ensuring strong viscoelastic properties [6].

AC is organised in different zones that can be divided into superficial, middle, deep and calcified cartilage zones [1,7]. Within each zone the abundant ECM can be subdivided into pericellular, territorial, and interterritorial sub-compartments [4,8].

Each zone shows a specific arrangement of COL II-containing fibres, thereby determining its mechanical properties [7]. In addition to the high stability of the ECM, the pericellular matrix (PCM) is particularly important for the transmission of mechanical stimuli [9,10]. PGs are responsible for osmotic properties, retain a lot of water, and are therefore important in resisting compressive forces. The content of PGs in the ECM increases from the superficial to the deep cartilage zones, where they are tightly packed and linked with collagen fibres to form a dense matrix [6,7]. In addition to COL II and PGs, other collagen types (e.g., collagen types IV, VI, IX, X, XI, XII, XIV), glycoproteins, and small non-collagenous proteins also play important roles in the structure and function of AC [1,11]. Several hundred proteins, some with molecular functions that are still unclear, have been identified in different zones of cartilage tissue. Recently, the use of proteomic techniques has provided important additional insight into the molecular composition of AC, which is becoming increasingly relevant for cartilage regeneration and tissue engineering approaches [12,13].

Besides the specialised architecture of the ECM, the cartilage thickness is different at different anatomical localisations in the joint to adapt to mechanical loads [14,15]. Regions exposed to higher mechanical loads exhibit increased cartilage thickness compared to areas under lower loading conditions [14,15,16]. However, it is not yet known whether the distribution of individual ECM proteins depends on the localisation in the joint and thus, in turn, on mechanical loading.

The precise composition of AC is crucial for its functionality and any alteration in the organisation of the ECM can lead to degenerative diseases such as OA [4,17]. It is well known that both COL II and PGs are degraded by proteases such as matrix metalloproteinases (MMPs) and A disintegrin and metalloprotease with thrombospondin-1-like domains (ADAMTS) during progression of OA [4,6,8,18]. In addition, the regulation of ECM proteins, proteases, and inflammatory factors are altered during OA, resulting in further degeneration of the tissue [8,19,20].

Cartilage has a low repair capacity and, therefore, regenerative and tissue engineering approaches have had some success counteracting cartilage loss during OA progression [21,22]. Tissue engineering and regenerative medicine are becoming increasingly important, both in the context of cartilage/bone regeneration and in the broader field of biomedical research [23,24]. However, in order to regenerate the tissue in the best possible way, it is crucial to break down the exact composition of the ECM. A deeper investigation into the individual matrix components and their distribution is essential for developing therapeutic approaches to treat cartilage injuries and diseases.

The present study focuses on a systematic analysis of healthy porcine AC at different anatomical localisations in the knee joint with special regard to cartilage thickness and localisation of specific ECM components. Due to its genomic similarity to humans, porcine tissue is commonly used in in vivo and in vitro experiments, providing high translational relevance [25,26].

## 2. Results

### 2.1. Cartilage Thickness and Structure

To investigate regional differences in cartilage morphology, histological analyses were performed on femoral condyles at defined positions of both the lateral (L) and medial (M) compartment. To obtain an overview of the tissue structure, hematoxylin and eosin (H&E) staining was performed on three individual knee joints (n = 3). Based on the study by Ma et al. (2024), we conducted a power analysis (one-tailed paired *t*-test; correlation = 0.3; α = 0.05; β = 0.1), which indicated a required sample size of n = 20 [27]. As sufficient specimens were available, the sample size was increased to 33, providing a calculated power of 100%. (Figure 1).

A schematic overview of the analysed positions on the femoral cartilage surface is shown in Figure 1b, with medial positions labelled M1 to M4 and lateral positions labelled L1 to L4, arranged from posterior to anterior.

Representative H&E-stained sections (n = 3) from the eight sampled locations reveal clear differences in cartilage thickness between the medial and lateral compartments (Figure 1a). The medial sections (M1–M4) show visibly thicker cartilage layers, characterised by a well-organised arrangement of chondrocytes and pronounced zonal architecture. The staining intensity is relatively homogeneous across all layers, with the exception of calcified cartilage and subchondral bone, where the staining is more intense. In contrast, the lateral sections (L1–L4) display a noticeably thinner cartilage layer, although the architecture and the overall staining intensity is similar. However, it can be observed that the staining in the middle cartilage layer, above the cell clusters, is more intense in the lateral compartment compared with the medial sections.

The average cartilage thickness in millimetres (n = 33) for each position, along with mean values for the medial and lateral compartments, is summarised in Figure 1c. Application of the linear mixed-effects model showed that cartilage on the medial side was on average 0.68 mm thicker compared to the lateral side (β = −0.678, SE = 0.068, t(149) = −9.93, *p* < 0.001, 95% CI [−0.813, −0.543]), which is in line with the overview staining.

Box plots illustrating the distribution of cartilage thickness for each of the eight locations are presented in Figure 1d. Each box (n = 33) represents the interquartile range, with the median indicated by a horizontal line, the mean marked by a cross, and individual outliers shown as dots. The medial positions (M1–M4) consistently exhibit higher median and mean cartilage thickness compared to the lateral positions (L1–L4). Among the medial regions, M2 and M3 show the highest thickness values, while in the lateral compartment, all positions (L1–L4) show relatively similar and lower thickness, with slightly greater variability observed at L1 and L2. Measurement site did not have a significant effect (β = −0.0064, *p* = 0.667), indicating that cartilage thickness did not systematically vary across the four positions within each side. Thus, lateral cartilage was consistently thinner than medial cartilage, independent of measurement site.

### 2.2. Proteoglycan and Collagen Type II Distribution

In addition to cartilage thickness, the composition of the ECM plays a critical role in maintaining the functional properties of AC [1]. To characterise the regional distribution of matrix components, histological and immunohistochemical staining was performed for PGs (Figure 2a) and COL II (Figure 2b) at eight anatomically defined positions (Figure 1b) of the knee joint.

Representative toluidine blue staining (n = 3), which selectively binds PGs (purple-blue), is shown in Figure 2a, with fast green used as a counterstain for non-cartilaginous structures. All eight positions show an increase in PG content from the superficial zone to the deeper zones, with strongest staining at the tidemark and calcified cartilage. In lateral samples, the deep cartilage zones in the region of the cell clusters show reduced PG content in comparison with the medial compartment. In lateral samples, PCM staining in the superficial and middle cartilage layers is visibly reduced, especially at positions L2 and L3. A similar, though less pronounced, pattern is observed in the medial samples.

Figure 2b illustrated a representative immunohistochemical staining for COL II (n = 3). In contrast to PG staining, COL II displays an inverse distribution, with the strongest red fluorescence signal in the superficial and middle zones, gradually decreasing toward the deeper cartilage layers. The cell clusters in the deep cartilage zone show increased staining intensity compared to the rest of the deep areas and therefore a higher amount of COL II. All in all, the cartilage tissue from the medial femoral condyle shows a higher intensity for the COL II staining compared to the lateral tissue.

In order to analyse whether these observations can also be quantitatively represented, proteome analyses of neighbouring tissue sections were performed.

### 2.3. Regulation of Proteins in Proteome Analyses

Proteomic analyses were conducted on cartilage tissue from three individual knee joints (n = 3) collected from eight distinct anatomical locations (Figure 1b). Figure 3 presents two heatmaps illustrating region-specific protein expression patterns based on z-transformed mean values of the three samples. In both heatmaps, proteins were clustered according to similar expression trends across the sampled regions, with the line plots in the upper panels summarising the average expression profiles of the respective clusters.

The proteins listed next to the curve exhibit the same trend, and only those without missing data are displayed in the heatmap below (Figure 3).

In Figure 3a, the included proteins are Collagen Type XI alpha 1 (COL11A1), Dystrophin (DMD), Eosinophil peroxidase (EPX), Galectin-13 (LGALS13), Non-secretory ribonuclease (LOC102163838), Lactotransferrin (LTF), Myeloperoxidase (MPO), Porcine myeloid antimicrobial peptide-23 (PMAP-23), Resistin (RETN), and Syntaxin-4 (STX4). The heatmap reveals consistently higher expression levels of these proteins in the medial positions M1 to M4, as indicated by the dark red coloration. Expression levels decline sharply in the lateral positions, with the lowest values typically observed at L3 and L4 and highest values at L2. Hierarchical clustering assigns these proteins to a single expression group with a characteristic medial-to-lateral decreasing trend, which is also reflected in the mean expression curve shown above the heatmap.

In Figure 3b, a different cluster of proteins is shown, including Aldehyde Dehydrogenase 18 Family Member A1 (ALDH18A1), ADP-Ribosylation Factor 4 (ARF4), Complement Factor D (CFD), Chitinase-1 (CHIT1), Alpha-L-Fucosidase 1 (FUCA1), Hexokinase-1 (HK1), Lipase Maturation Factor 2 (LMF2), Matrin-3 (MATR3 Member RAS Oncogene Family (RAB2A), and Ribosomal Protein S21 (RPS21). These proteins display an inverse pattern to those in Figure 3a. Expression levels are low in the medial samples M1 to M4, as indicated by blue shading, and increase toward the lateral regions, with the highest values observed at L4. The average expression curve above the heatmap confirms a gradual medial-to-lateral increase in abundance for this protein group.

The proteins shown in Figure 3 are mainly involved in cellular processes such as inflammatory and immune responses, metabolism, enzymatic functions, or signal transduction.

To assess whether ECM-related proteins were differentially regulated, the data for significantly changed proteins (FDR < 0.05) were analysed for enrichment of relevant Gene Ontology (GO) terms and evaluated for statistical significance. The results demonstrated that the GO terms ‘collagen-containing extracellular matrix’ and ‘extracellular matrix’ were highly upregulated in the medial samples. In contrast, only one lateral position showed an upregulation of these terms, which was comparatively modest (Appendix A).

The following heatmap illustrates normalised (z-transformed) expression levels of ECM proteins, chosen based on their established roles in cartilage homeostasis and disease. The analysed proteins are Collagen Type II alpha 1 (COL2A1), Aggrecan (ACAN), Cartilage Oligomeric Matrix Protein (COMP), Fibrillin-1 (FBN1), Decorin (DCN), and Matrilin-3 (MATN3), all of which play critical roles in cartilage structure and function (Figure 4).

Hierarchical clustering of the proteins revealed three major expression clusters. The first cluster comprises COL2A1, ACAN, COMP, and FBN1, which share a common expression pattern characterised by upregulation in the medial and downregulation in the lateral regions.

DCN, assigned to cluster two, exhibits a more variable expression profile, with the highest relative abundance at position M4 and the lowest at L2. The remaining medial and lateral samples display intermediate expression levels, following an ascending trend from posterior to anterior positions in the medial condyle (M1 to M4).

In contrast to DCN, an inverse distribution pattern is observed for MATN3 (cluster 3). Within the medial samples, its abundance decreases progressively from posterior to anterior regions. Among the lateral samples, no consistent pattern emerges, with protein levels displaying considerable variability.

However, among these clearly discernible differences, only the following exhibit statistically significant variations in protein abundance (*q*-value < 0.01): COL2A1: M2/M3 to L2; ACAN: M2 to L2/L4, M3 to L2/L4; COMP: M2 to L2; FBN1: M2/M3 to L4; DCN: M3 to L2; MATN3: no significant deviations. The whole dataset can be found in Appendix A.

### 2.4. Organisation of ECM Proteins

In order to verify the results of the proteome analysis, cartilage tissue from three individual knee joints (n = 3) was homogenised and dissolved in order to detect individual proteins using Western blot. Positions M2 and L2 were selected as representative opposing sites of the joint, as they most frequently exhibited significant differences in the proteome. The results presented in Figure 5 clearly confirm the proteomic data for matrilin-3, decorin, and COMP. However, the antibody for fibrillin-1 unfortunately did not work for the Western blot.

Figure 5 presents the Western blot results, showing both the band patterns and the normalised band intensities in graphical form. Quantitative proteomic analysis indicated equivalent expression levels of matrilin-3 at positions L2 and M2. This was further supported by Western blot detection of the 55 kDa monomer, which showed comparable signal intensities (L2 = 1.0; M2 = 1.06) in both samples with no significant deviation (*p* = 0.62).

For both decorin and COMP, markedly stronger bands were observed in sample M2 compared with L2, validating the proteomic data. Multiple bands were detected for each protein across various molecular weight ranges: decorin appeared predominantly between 180 and 35 kDa, with particularly intense signals at approximately 110 and 45 kDa. Most likely, the antibody detects differentially glycosylated forms of decorin and the core protein that has a predicted molecular weight of 45 kDa [28]. Although the mean band intensity of M2 (1.81) was markedly higher compared to L2, the number of biological replicates for this protein was too low to yield a statistically significant result, despite the *p*-value (0.052) being close to the significance threshold.

Intact COMP would run at around 100 kDa, while smaller bands most likely detect degradation products that have been described previously [29]. For COMP, higher molecular weight bands exhibited greater intensity, whereas lower molecular weight fragments were present as weaker bands. COMP levels were significantly higher in M2 compared to L2 (*p* = 0.0051).

To assess not only the quantitative differences but also the zonal distribution of ECM proteins, immunohistochemical staining was performed (Figure 6).

Figure 6 illustrates representative immunohistochemical staining of three individual knee joints (n = 3) of matrilin-3, decorin, COMP, and fibrillin-1 at the localisations L2 and M2, visualised by green fluorescence.

The primary antibodies used for staining were developed based on human protein sequences. To ensure that they also detect porcine proteins, the human and porcine sequences were compared. All analysed proteins exhibited a very high proportion of identical amino acid sequences, making it highly likely that the antibodies also bind to the corresponding porcine proteins: Decorin = 91.1%; Matrilin-3 = 88.2%; Fibrillin-1 = 96.9%; Collagen type II = 97.7%, and COMP = 93.3% (Appendix A).

Immunohistochemical staining for matrilin-3 indicates distinct differences in spatial distribution. In the lateral sample, matrilin-3 was absent in the superficial zone, whereas M2 shows clear staining in this region. Both localisations exhibit strong signals in the middle zone, while signal intensity decreases in the deep zone, especially in M2. Although overall matrilin-3 levels were comparable, the distribution pattern varied notably.

The overall intensity of decorin staining was relatively low; however, a markedly increased signal was observed at position M2 in comparison to L2. Notably, decorin appeared to align along a fibrous structure, particularly prominent in the superficial zone and the middle cartilage layer. At position M2, this specific arrangement extended significantly deeper into the tissue compared to other regions.

The overall staining intensity of COMP is low, with ECM staining restricted to the superficial layer and upper cartilage zones. In the middle and deeper zones, COMP is predominantly detected in a cellular and pericellular pattern.

Fibrillin-1 staining clearly revealed individual fibrils, which were strongly expressed in both the superficial and middle zones of the cartilage. In the deep cartilage layers, those fibrils were no longer detectable.

## 3. Discussion

To investigate regional differences in cartilage composition, proteomic analyses, Western blotting, and histological/immunohistochemical staining were performed on porcine AC from anatomically distinct regions. It is well-established that mechanical loading significantly influences cartilage tissue, affecting chondrocyte metabolism and cartilage thickness [14,15,16]. We hypothesised that AC also undergoes adaptive changes in ECM protein composition that vary by anatomical location.

In this study, porcine tissue of unknown sex was used. Several studies have investigated differences between male and female human cartilage, showing variations in cartilage thickness and joint size [30,31]. Furthermore, sex-specific differences in phenotype and progression of OA have been reported; however, additional factors such as age, BMI, and lifestyle are also highly relevant. These studies do not indicate substantial differences in cartilage composition between sexes [32,33,34]. Our results show consistent cartilage thickness across 33 porcine joints with low variance, suggesting that sex does not have a major impact in this context. Despite the unknown sex of the animals, significant results were obtained. Nevertheless, this study is limited by the fact that sex-specific effects cannot be excluded, which should be considered in future research. Porcine tissue is frequently used in OA research [35,36,37] because both the joint size and the mechanical loading conditions closely resemble those of humans [38,39]. Due to the limited availability and heterogeneity of healthy human samples, porcine cartilage was used for molecular analysis, as it is closely similar to human cartilage in terms of cartilage thickness, collagen fibre orientation, and structural organisation [38,39,40]. A review by Cone et al. (2017) summarised various porcine studies and, despite some limitations, concludes that porcine models are gaining importance as translational models in musculoskeletal tissue engineering and regenerative medicine, and are also well-suited to address research questions in the field of biomechanics [41]. Porcine models are also suitable for modelling the human OA phenotype due to their high reproducibility [42,43]. The biomechanical properties of healthy human and porcine cartilage were compared, concluding that, although differences in loss angle represent a limitation, porcine cartilage remains a suitable model for mechanical studies, including tissue engineering applications, particularly regarding dynamic modulus and dynamic stiffness [44].

In this study, the medial condyle showed significantly greater cartilage thickness than the lateral condyle (β = −0.678, SE = 0.068, t(149) = –9.93, *p* < 0.001, 95% CI [−0.813, −0.543]) (Figure 1b–d), aligning with human values and suggesting differential mechanical loading as a contributing factor [38,39,40]. The medial compartment is typically subjected to higher mechanical loads [14,45,46], and because most pigs exhibit neutral limb alignment [47], it is very likely that thickness differences are load-induced rather than anatomical. Furthermore, the thickness measurements revealed no significant differences within a single condyle (β = −0.0064, *p* = 0.667). Two studies have shown that higher forces are typically transmitted through the medial compartment of the human knee during gait [48,49]. Porcine knee joints were analysed using T2 (transverse relaxation time) mapping and it was demonstrated that higher T2 values occur in the lateral condyle under loading. In contrast, the medial condyle, which is generally subjected to higher loads, exhibits a denser collagen network, resulting in lower T2 values [50].

While cartilage thickness did not differ significantly between positions 1–4, previous studies indicate regional variations in load distribution in human AC [45,46]. Accordingly, positions M2/L2 and M3/L3 in our model are likely subject to higher loads.

Histological analysis (Figure 1a) confirmed a zonal architecture in porcine cartilage consistent with human tissue, including distinguishable superficial, middle, deep, and calcified layers, as well as the pericellular (PCM) and interterritorial matrix [1,4,7,8]. Despite prior research addressing zonal and disease-related differences in ECM composition [12,51,52,53], topographical heterogeneity remains underexplored. This study specifically targeted the femoral cartilage, which is more frequently affected by degenerative conditions such as OA [53,54].

Proteomic clustering revealed distinct medial–lateral differences, particularly in proteins associated with inflammation, metabolism, enzymatic activity, and signal transduction (Figure 3). GO term analysis identified significant enrichment of “extracellular matrix” and “collagen-containing extracellular matrix” proteins in the medial compartment, indicating a denser ECM and supporting the proposed relationship between mechanical load and ECM composition.

Besides COL II and PGs as the main ECM components, we focused on COMP, decorin, matrilin-3, and fibrillin-1. While the role of fibrillin-1 in cartilage and OA is not yet fully understood [55,56], matrilin-3 [57,58], decorin [59], and COMP [29,60] are known to play a role in OA progression and are therefore of great interest. Other proteins such as matrilin-1 and -4, or collagens type IX, X, XII, and XIV, could not be included due to the lack of suitable antibodies, which represents a limitation of this study.

Major ECM proteins such as PGs and COL II displayed region-specific distribution patterns consistent with prior literature [6,7]. PGs increased from the superficial to the calcified zone (Figure 2a), while COL II showed the inverse trend (Figure 2b). Both were more abundant in medial cartilage, as confirmed by proteomic analysis (Figure 4) and immunostaining (Figure 2), suggesting enhanced viscoelasticity and structural resilience in response to mechanical loading [1,5,6,7].

Proteomic profiling identified an overall higher expression of matrilin-3, decorin, COMP, and fibrillin-1 in the medial condyle, with significant deviations (*p* < 0.05 for each, except for matrilin-3) for some localisations (Figure 4). Western blot and immunohistochemical analyses of representative sites (L2 and M2) corroborated these findings.

Matrilin-3 was localised in the superficial zone only in medial samples (Figure 6), indicating increased ECM crosslinking capacity in this area, and consequently, a higher capacity to withstand mechanical loading [57,61,62]. The overall staining intensity showed no visible differences between L2 and M2, which is consistent with the proteomic profile (Figure 4) and the Western blot detection (Figure 5).

COMP and decorin were found to be markedly higher in medial cartilage by proteomic and Western blot analyses (Figure 4 and Figure 5), with decorin also showing greater staining intensity medially (Figure 6). As a linker-protein, decorin regulates aggrecan–collagen interactions, contributing to tissue integrity and load-bearing capacity [63,64], while COMP is a mechanosensitive protein facilitating ECM–cell interactions [65,66,67]. The absence of zonal differences in the distribution of COMP and decorin suggests that, in contrast to matrilin-3, it is not their spatial localisation but rather their overall abundance that may contribute to an increased load-bearing capacity of the tissue.

Fibrillin-1, a core component of microfibrils, was also enriched in the medial compartment (Figure 4). Its structural role includes mechanical support and regulation of growth factor availability, notably transforming growth factor-β [55,68,69,70,71]. The increased presence of fibrillin-1 suggests enhanced mechanical competence of the medial ECM. Immunohistochemical staining of fibrillin-1 (Figure 6) revealed organised fibrillar structures surrounding chondrocytes, emphasising its role in providing mechanical support and maintaining ECM integrity.

Overall, these results suggest that higher quantities of several ECM proteins in the medial joint compartment may contribute to its enhanced ability to withstand locally increased mechanical loads, thereby maintaining cartilage stability and integrity. The present analyses align with a previous study, which reported that differences between the medial and lateral compartments occur in porcine knee joints, and are consistent with the assumption that medial cartilage may be more densely structured to withstand higher mechanical loads [50].

This supports the importance of regional variations, as emphasised by a recent study, which identified it as a critical factor in models of joint injury and OA [72].

Methodologically, proteomic and Western blot analyses offered high sensitivity for quantifying protein differences, which staining procedures did not always consistently match (e.g., COMP, fibrillin-1), possibly due to tissue accessibility or paraffin embedding effects. However, staining remains valuable for elucidating protein distribution patterns and zonal variations (e.g., matrilin-3, PGs, COL II). Study limitations include the use of immature porcine tissue (6–7 months old), which may not fully represent mature adult cartilage [39].

In summary, this study highlights pronounced regional heterogeneity in cartilage ECM, with the medial compartment showing a molecular profile indicative of higher ECM protein expression, thus a denser tissue network. These findings could be highly relevant for tissue engineering and cartilage replacement therapies in OA, suggesting that considering regional differences and matching donor sites could lead to more successful regenerative approaches. Subsequent analyses placing these results in the context of OA could provide valuable insights for the development of more targeted and effective treatment strategies.

## 4. Materials and Methods

### 4.1. Sampling of Osteochondral Cylinders

Osteochondral explants (Figure 1b) were harvested from porcine knee joints within 24 h post-mortem. The joints of German domestic pigs (*Sus scrofa*) occur as slaughterhouse waste in the food industry. The pigs weighed approximately 130 kg and were 6–7 months old at the time of slaughter. Information on the sex of the animals was not available.

The joints were macroscopically examined for signs of disease and only intact, healthy joints were used for experiments. Osteochondral cylindric specimens with a diameter of 8 mm were harvested from the femoral articular surface, using a drill and a specialised tool (LifeTec Group, Eindhoven, The Netherlands). Eight samples per femur were taken at anatomically defined localisations, four on the medial (M1–M4) and four on the lateral (L1–L4) femoral condyle (Figure 1b).

### 4.2. Sample Preparation

The tissue explants were transferred to 4% paraformaldehyde in PBS (pH 7.4; Morphisto, Frankfurt, Germany) directly after sampling and incubated at 37 °C and 100 rpm for 24 h. After fixation, the bone was decalcified in 10% formic acid solution (reagent grade ≥ 98%, Carl Roth, Karlsruhe, Germany) for about 14 days at 37 °C and 100 rpm, until the bone was soft enough to cut. The samples were cut in half lengthwise, washed under running water for one hour and were stored at 4 °C in 70% ethanol until embedding.

The tissue explants were prepared for paraffin embedding by automatic dehydration. Tissue sections of 5 µm thickness were prepared with the Leica RM2235 microtome (Leica Biosystems, Nussloch, Germany), pulled onto microscope slides and stretched at 42 °C. After that, the slides were dried at 37 °C for at least 24 h before being stored at room temperature.

For histological and immunohistochemical staining, as well as proteomic analysis, tissue sections from the exact same animals were used. The sections were deparaffinised, by melting the paraffin (Surgipath Paraplast Plus, Leica Biosystems, Germany) at 60 °C, followed by two changes in xylene (histological grade, Sigma Aldrich, Hamburg, Germany) and a descending alcohol series from 100% to 70% ethanol (Ethanol 99.5% denatured with 1% MEK; VWR, Darmstadt, Germany). Finally, the sections were rehydrated in deionised water. Each individual step was incubated for at least five minutes.

### 4.3. Histology and Immunohistochemistry

For H&E staining, the rehydrated tissue sections were soaked in Hematoxylin (Mayer’s hematoxylin solution, 254766.1211; PanReac AppliChem, ITW Reagents, Germany) for 10 min and then washed with tap water, soaked in 70% ethanol for 10 sec, and incubated in tap water for 10–15 min. Per 100 mL Eosin dye (Eosin G-solution 0.5% water-based, X883, Carl Roth, Germany) one drop of glacial acetic acid (Rotipuran 100%, Carl Roth, Germany) was added. The sections were soaked in the solution for 15 min and subsequently washed in tap water again.

Toluidine blue was used for the staining of PGs in the cartilage tissue, together with fast green for counterstaining. After rehydration in deionised water, the slides were transferred directly into a 0.04% toluidine blue solution (toluidine blue, CAS# 6586-04-5, Sigma Aldrich, Germany) in 0.1 M sodium acetate buffer and stained for ten minutes in the dark. The sections were then washed in three changes in deionised water before counterstaining with 0.02% fast green (fast green FCF, CAS# 2353-45-9, Sigma Aldrich, Germany) in deionised water for three minutes. The dye was removed by three changes in deionised water.

After histological staining, the tissue sections were dehydrated briefly in two changes each of 96% and 100% ethanol, followed by two changes in xylene. Using coverslips and mounting medium containing xylene, the slides were covered and stored at room temperature until microscopic analyses.

For immunohistochemical staining, the rehydrated sections were pretreated prior to staining. Pepsin (1 mg/mL in 0.5 M acetic acid, 45 min at 37 °C; Sigma Aldrich, Germany) was used for the detection of COL II, and hyaluronidase (250 U/mL in hyaluronidase buffer (100 mM NaH_2_PO_4_, 100 mM sodium acetate, pH 5), 40 min at 37 °C; Carl Roth, Germany), for all other proteins. Prior to hyaluronidase treatment, the tissue was acidified for five minutes with hyaluronidase buffer.

After two washes with deionised water, the sections were blocked with 10% foetal bovine serum (FBS) in PBST (PBS + 0.1% Tween-20) for one hour at 37 °C. The blocking solution was then removed, and primary antibodies were applied. All primary antibodies were prepared in 1% FBS in PBST, incubated overnight at 4 °C, and diluted as listed in Table 1.

**Table 1 ijms-26-09331-t001:** Antibodies used for immunohistochemical staining.

Target	Dilution	Host	Source
h-Collagen type II	1:500	Mouse	Anti-Collagen Type II (Ab-1) Mouse mAb (II-4C11), CP18, Merck, Germany
h-Matrilin-3	1:1000	Rabbit	Klatt et al. 2000 [73]
h-COMP	1:1000	Rabbit	DiCesare et al. 1994 [74]
h-Fibrillin-1	1:2000	Rabbit	Morcos et al. 2022 [75]
h-Decorin	1:1000	Rabbit	Kupka et al. 2020 [76]

The listed antibodies are based on human protein sequences, which does not preclude their applicability to proteins of other species. According to the datasheet, the collagen type II antibody detects not only human protein but also rat, bovine, and rabbit. To ensure that all antibodies were suitable for porcine samples, the protein sequences were retrieved from the https://www.uniprot.org/ database (accessed on 9 September 2025) and aligned using the NIH BLASTp (https://blast.ncbi.nlm.nih.gov/Blast.cgi) (accessed on 9 September 2025) to determine their homology.

The primary antibody solutions were removed by three changes in PBST before adding the secondary antibodies (Alexa Fluor 594 goat anti-mouse IgG (H+L), 2 mg/mL and Alexa Fluor 488 goat anti-rabbit IgG (H+L), 2 mg/mL, Invitrogen, Germany). Both secondary antibodies were diluted 1:500 in 1% FBS in PBST and incubated for 45 min at 37 °C. The slides were washed with PBST and one final wash with deionised water before the tissue sections were mounted using Fluoromount-G (Invitrogen by Thermo Fisher Sientific, Dreieich, Germany) and coverslips.

All microscopic analyses were carried out with the Keyence fluorescence microscope BZ-X810 and were edited with the Keyence BZ-X800 analyser software V 1.1.2.4 (Keyence, Neu-Isenburg, Germany).

### 4.4. Determination of Cartilage Thickness

The cartilage thickness was determined by using microscopic overview images of the tissue sections. Using ImageJ (V 1.54d), the thickness was measured from the cartilage surface to the cartilage/bone interface. The results for each localisation were based on 33 different knee joints. For each knee joint and localisation (M1–M4 and L1–L4) one tissue section was selected for thickness measurements. An average of ten individual measurements was used for each tissue section. A linear mixed-effects model was fitted with the animal as a random effect and side (medial vs. lateral) and measurement site (positions 1–4) as fixed effects.

### 4.5. Proteomic Analysis

Eight sections from porcine cartilage (n = 3) were lysed with 4% SDS in PBS, 15 mM CAA, and 5 mM TCEP, heated to 95 °C for 60 min, and sonicated using the bioruptor (Bioruptor Pico, Diagenode, Seraing, Belgium) with 10 cycles of 30 s and 30 s break. Protein digestion was performed following the Single-Pot Solid-Phase-enhanced Sample Preparation (SP3) protocol [77]. In brief, both hydrophilic and hydrophobic beads were added to the sample and bound by adding 1:1 volume of acetonitrile (ACN). After 8 min incubation time, magnetic beads were immobilised and washed 2× with 70% ethanol and ACN. Proteins were digested with trypsin (substrate:enzyme ratio 100:1) overnight at room temperature. Samples were acidified by using 100 µL 0.1% formic acid (FA) followed by a clean-up with house-made SDB-RPS tips. Desalted peptide separation was performed on a nanoElute HPLC system (Bruker, Germany) equipped with 15 cm PepSep columns (EvoSep, Odense, Denmark) at 45 °C with a gradient length of 60 min. Mobile phases were composed of 0.1% FA as solvent A and 0.1% FA in ACN as solvent B. The HPLC system was coupled to a timsTOF pro 2 using a CaptiveSpray source (both Bruker, Hanau, Germany). Samples were measured in dia-PASEF mode with ion mobility calibrated using three ions of Agilent ESI-Low Tuning Mix following vendor specifications. The dia-PASEF window was ranging in dimension 1/k0 0.7–1.45, with 16 × 2 Th windows, and in dimension *m*/*z* from 350 to 1250.

The mass spectrometry proteomics data have been deposited to the ProteomeXchange Consortium via the PRIDE partner repository with the dataset identifier PXD068032 [78]. Acquired spectra were analysed with DIA-NN (V1.8.1) using library free search against UniProt Oryctolagus cuniculus (*Sus scrofa*) database (August, 2024) [79]. Mass ranges were set according to the settings of the mass spectrometer. Data was further processed using R (V 4.2.2), with the libraries tidyverse, diann, data.table, magrittr, FactoMineR, factoextra and ggplot2, gprofiler, and ggplot2. Data input was filtered for unique peptides, *q*-Value < 0.01, Lib.Q.Value < 0.01, PG.Q.Value < 0.01, Global.Q.Value < 0.01, Quantity.Quality > 0.7, Fragment.count ≥ 4.

Quality control of iRT peptides was performed in Skyline-Daily (V 22.21.391). Further calculations were performed in R using the packages openxlsx, vsn, limma, corrplot, ggplot2, gprofiler, and statmod.

Further analyses were performed in perseus (V 1.6.5.0) and InstantClue (V 0.10.10.20211105). ANOVA analyses and unpaired Student’s *t*-tests were carried out with s0 = 0.1, permutation-based FDR < 0.05, and 500 randomisations. Gene Ontology term enrichment was performed on the highest-abundance proteins per group using g:Profiler (g:SCS threshold < 0.05). To assess directional trends, we additionally applied Pearson correlation. ANOVA was used to test for general differences between groups, whereas *t*-tests were used for specific unpaired comparisons. Data values in tables are log_2_-transformed, and values shown in heatmaps are z-score normalised.

### 4.6. Protein Isolation from Cartilage Tissue

For the isolation of proteins from cartilage tissue, osteochondral explants were harvested as described above. The cartilage was separated from the bone and transferred to a reaction tube with 20 µL extraction buffer (4 M guanidine-HCl, 50 mM tris base, 10 mM EDTA, pH 7.2) per mg cartilage. The tissue was mechanically minced and then homogenised using a dispersing unit (Polytron PT 1200 E, Polytron, Germany and Dispersing aggregate, Ø 5 mm, Kinematika, Eschbach, Germany). The tubes were incubated overnight at 4 °C on a shaking incubator. The next day, the soluble protein fraction was separated from the insoluble part by centrifugation (10 min, 14,600 rpm). The supernatants contained the soluble proteins, which were used for protein precipitation.

For protein precipitation, four times the amount of acetone (≥ 99.5% for synthesis, Carl Roth, Germany) was added in proportion to the sample and incubated overnight at −20 °C. The proteins were pelleted by centrifugation (10 min, 14,600 rpm) and the pellets were washed in a PBS/acetone (1:10) mixture for two hours and then centrifuged again. Supernatant was discarded and the remaining acetone was allowed to evaporate for approx. 5 min, before the pellet was resuspended in loading buffer (1.5 times the amount of protein extract) and then denatured for 10 min at 70 °C.

### 4.7. Western Blotting

The protein distribution in the cartilage extracts of localisations L2 and M2 was analysed by Western blotting. A 12% acrylamide gel was used to separate the isolated proteins, which were then blotted onto a PVDF membrane (0.2 µm; Thermo Scientific, Germany) using Towbin transfer buffer (25 mM tris, 192 mM glycine, 10% (*v*/*v*) methanol, and 0.01% (*w*/*v*) SDS) for 60 min in a semi-dry process.

Prior to antibody incubation, the membranes were blocked for one hour in 5% skimmed milk in PBST. The antibodies (details and dilutions in Table 1) were also prepared in 5% skimmed milk and were incubated overnight at 4 °C. After three washing steps with PBST, the secondary antibody (1:1000 in PBST; goat anti-rabbit IgG (H+L), HRP conjugated, 0.01 mg/mL, Invitrogen, Germany) was incubated for 45 min at room temperature.

For chemiluminescence detection, the SuperSignal West Atto Ultimate Sensitivity Substrate solution (Thermo Fisher, Germany) was used. The two components, peroxidase solution and substrate solution, were mixed together in a ratio of 1:1, and then deionised water was added, also at a ratio of 1:1. After application to the membrane, the solution was allowed to soak in for one minute, before using the iBright 1500 system (Invitrogen, Germany) for chemiluminescence detection.

For semi-quantification of band intensity, values were normalised to position L2 (L2 = 1) using the iBright 1500 system. Statistical evaluation of the normalised data was performed with a one-sample *t*-test, statistical significance was set at *p* < 0.05.

## 5. Conclusions

This study demonstrates distinct regional variations in the ECM composition of porcine AC, which appears to be closely linked to mechanical loading patterns. The medial femoral condyle exhibited a consistently higher abundance of structural ECM proteins, suggesting an adaptive response to increased biomechanical loadings. Proteomic profiling, corroborated by Western blotting and immunohistochemistry, revealed region-specific enrichment of proteins involved in structural support, mechanotransduction, and matrix organisation (PGs, COL II, matrilin-3, decorin, COMP, and fibrillin-1).

These findings highlight the significance of anatomical location in cartilage biology and pathology, demonstrating that regional variations in ECM composition are influenced by local mechanical environments. Identifying such regional differences is critical for improving tissue engineering and chondrocyte/cartilage replacement therapies in diseases such as OA. Due to the similarities between porcine and human AC in ECM composition, anatomy, and loading, our findings may offer insights relevant for optimising OA treatment in humans. Nevertheless, certain limitations must be considered, especially with cross-species models, which are addressed in the discussion.

Treatments may achieve better outcomes when donor tissue is sourced from the same condyle, ensuring a closer match in ECM composition and biomechanical properties.

## Figures and Tables

**Figure 1 ijms-26-09331-f001:**
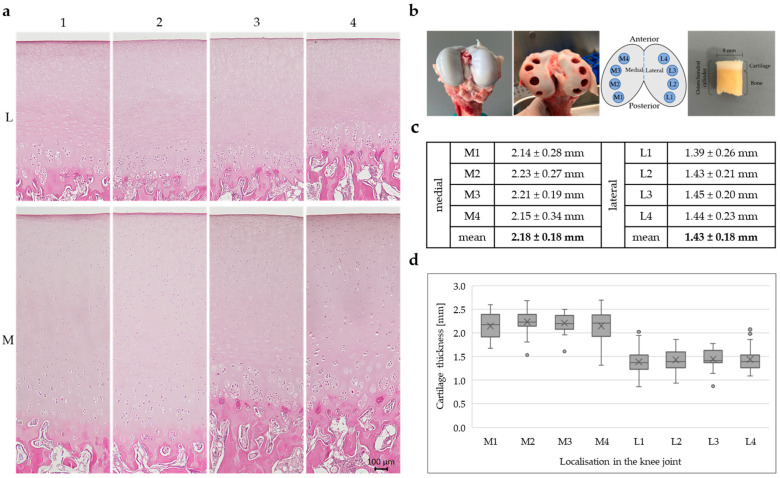
(**a**) Representative H&E staining of porcine cartilage tissue (n = 3) at eight anatomically defined localisations; scale is 100 µm, all microscopic images were taken with the Keyence fluorescence microscope BZ-X810. (**b**) Healthy porcine articular cartilage before and after sampling; sampling localisations, four medial (M1–M4) and four lateral (L1–L4) positions; porcine osteochondral cylinder with a diameter of 8 mm. (**c**) Cartilage thickness determined from cartilage sections of 33 individual knee joints (n = 33); statistical analysis was performed using a linear mixed-effect model: lateral cartilage thickness was 0.68 mm lower than medial thickness (β = −0.678, SE = 0.068, t(149) = −9.93, *p* < 0.001, 95% CI [−0.813, −0.543]). (**d**) Box plots show the interquartile range of cartilage thickness for all eight positions (n = 33), horizontal line: median, cross: mean value, dots: individual outliers.

**Figure 2 ijms-26-09331-f002:**
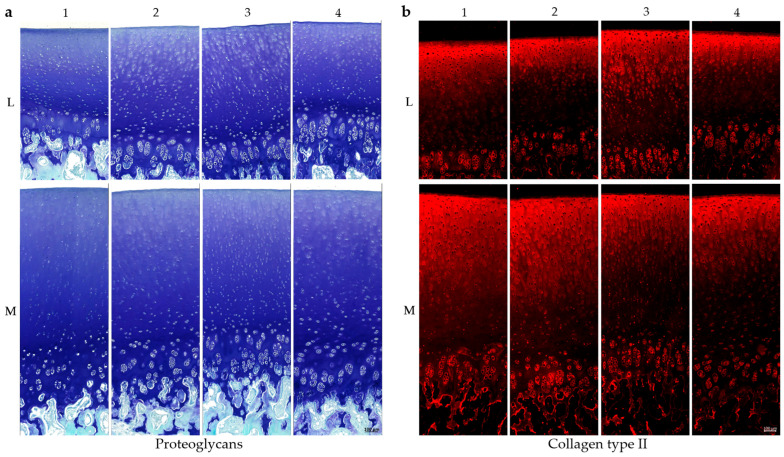
Histological and immunohistochemical staining of porcine cartilage tissue at eight anatomically defined localisations, M1–M4: medial samples, L1–L4: lateral samples. (**a**) Staining for proteoglycans with toluidine blue and counterstaining with fast green. (**b**) Staining for collagen type II (antibody details in Table 1, methods), secondary antibody: Alexa Fluor 594 goat anti-mouse (details in Section 4); scale is 100 µm. All microscopic images were taken with the Keyence fluorescence microscope BZ-X810.

**Figure 3 ijms-26-09331-f003:**
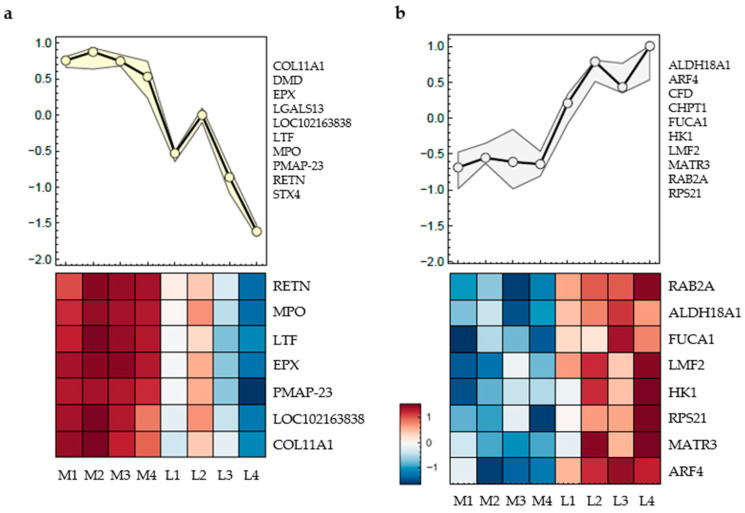
Proteomic profiles of porcine cartilage tissue (n = 3) at eight anatomically defined localisations, M1–M4: medial samples, L1–L4: lateral samples. The curves (upper part) represent the mean expression curve of all proteins in the cluster, the heatmaps include proteins without missing data; only proteins with significantly different regulation between positions are shown. *q*-value < 0.01 was considered statistically significant. (**a**) Cluster 1: Significantly regulated proteins following a medial-to-lateral decreasing trend; (**b**) Cluster 2: Significantly regulated proteins following a medial-to-lateral increasing trend. List of all proteins are given in the text below, heatmaps represent the mean value of three individual knee joints.

**Figure 4 ijms-26-09331-f004:**
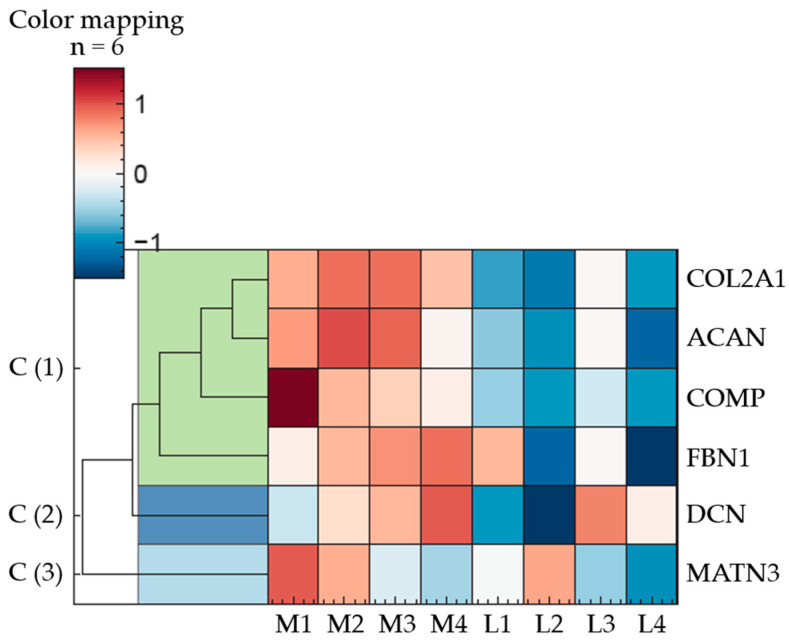
Regulation of ECM proteins in porcine articular cartilage (n = 3); eight anatomically defined localisations, M1–M4: medial samples, L1–L4: lateral samples. *q*-value < 0.01 was considered statistically significant. The heatmap illustrates ECM proteins in three expression clusters; Cluster 1: COL2A1, ACAN, COMP, FBN1; Cluster 2: DCN; Cluster 3: MATN3; description of the proteins are given in the text below, the heatmap represents the mean value of three individual knee joints.

**Figure 5 ijms-26-09331-f005:**
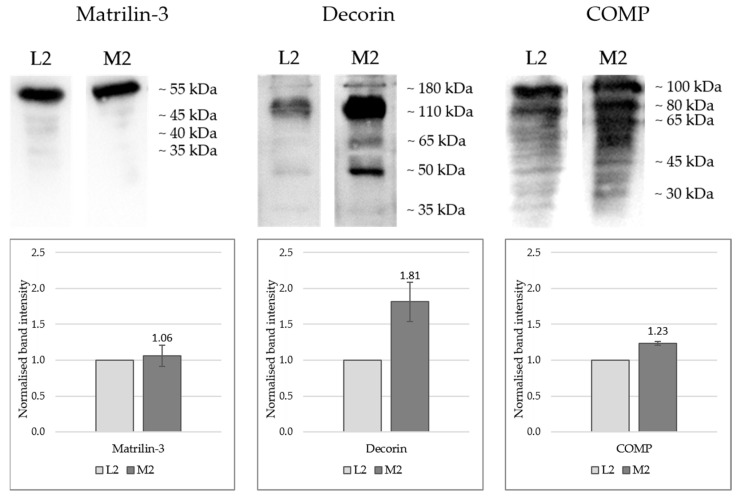
Western blot detection of matrilin-3, decorin, and COMP in protein extracts from porcine cartilage tissue from three individual knee joints (n = 3) at two anatomically defined localisations, L2: lateral sample, M2: medial samples. Proteins were extracted using a 4 M guanidine-HCl buffer followed by acetone precipitation and denaturation in loading buffer; SDS-PAGE was performed using 12% acrylamide gels, and protein transfer onto PVDF membranes with Towbin transfer buffer (described in the methods). Each lane was loaded with 15 µL of sample, representing the extract obtained from 500 µg of cartilage. Primary antibody information: see Table 1, methods; HRP-conjugated secondary antibody and SuperSignal West Atto Ultimate Sensitivity Substrate solution were used for chemiluminescence detection (see Section 4); Detection: iBright 1500 system, Invitrogen. One-sample *t*-tests were performed, statistical significance: *p* < 0.05 (COMP: *p* = 0.0051; Decorin: *p* = 0.052; Matrilin-3: *p* = 0.62).

**Figure 6 ijms-26-09331-f006:**
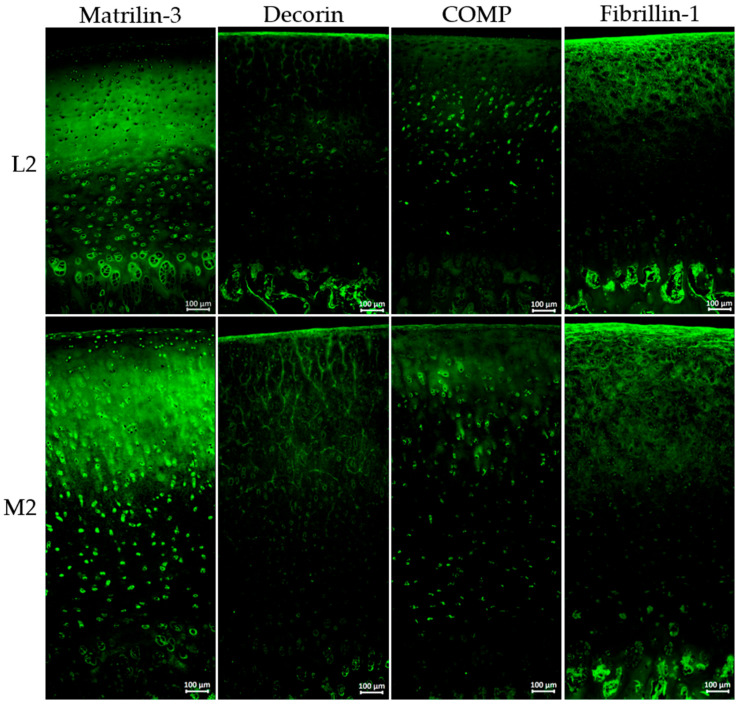
Immunohistochemical staining of porcine cartilage tissue at eight anatomically defined localisations, M1–M4: medial samples, L1–L4: lateral samples. Representative staining of three individual knee joints (n = 3); staining for matrilin-3, decorin, COMP, and fibrillin-1 (antibody details in Table 1, Section 4), secondary antibody: Alexa Fluor 488 goat anti-rabbit (details in Section 4); nuclear staining was not included due to autofluorescence in paraffin-embedded tissue sections; magnification for all images ×200; all microscopic images were taken with the Keyence fluorescence microscope BZ-X810.

## Data Availability

The raw data can be found in the Appendix A.

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
