# Peer review of "Localisation-Dependent Variations in Articular Cartilage ECM: Implications for Tissue Engineering and Cartilage Repair"

_ijms, 2025, doi:10.3390/ijms26199331_

Round 1
Reviewer 1 Report
Comments and Suggestions for Authors
The manuscript provides a comprehensive analysis of localisation-dependent variations in articular cartilage extracellular matrix in porcine knee joints, integrating conventional histology, immunohistochemistry, proteomics, and western blot techniques to characterize structural and molecular differences relevant for tissue engineering and cartilage repair.
While the manuscript is well written and is generally scientifically sound, I identified the following concerns.
- Were both male and female animals used in this study? If so, was sex a confounding variable? Please clarify the sex of the animal materials or comment on potential sex-specific differences in cartilage composition.
- Although porcine cartilage is used as an alternative model for human tissue, the extrapolation to human therapeutic strategies should be discussed with in a bit more detailed way. The authors should specifically address differences in ECM composition and mechanical loading patterns between porcine and human knees, with references to comparative studies. The limitations relating to species differences deserve expanded discussion in the conclusion and discussion sections.
- The statistical approach is generally well described, but the manuscript would benefit from inclusion of more detail about the selection and validation of statistical tests, effect sizes, and corrections for multiple comparisons (e.g., clearly explain use of FDR thresholds and ANOVA randomisations). Furthermore, I was wondering why did the authors only use proteins without missing data. I’m sure with proper and widely accepted imputation methods, they could have identified more (and potentially novel or exciting) proteins in this dataset. I was also wondering whether the authors performed principal component analysis (PCA) on the samples and how did (would) the samples cluster/separate on such a plot?
- For the western blot analysis, semi-quantification methodology (densitometry, normalization) should be performed and described in detail, rather than just visually compared.
- The criteria for selecting ECM proteins for clustering and detailed analysis (Figures 3 and 4) should be more explicitly justified. How many clusters were used? Why were these particular proteins (matrilin-3, decorin, COMP, fibrillin-1) chosen, and were other relevant ECM proteins investigated but excluded due to technical or biological reasons?
- The manuscript claims raw data is available in supplementary material; however, that was only an Excel export. The manuscript also states that MS data are available in the ProteomeXchange Consortium via the PRIDE partner repository. However, no accession number was given.
Author Response
Comment 1: Were both male and female animals used in this study? If so, was sex a confounding variable? Please clarify the sex of the animal materials or comment on potential sex-specific differences in cartilage composition.
Response 1: We thank the reviewer for raising this point. The sex of the animals used in this study is not known and was also not determined. This fact has now been explicitly stated in the Methods section of the revised manuscript “Information on the sex of the animals was not available.” (line: 460-461) We also clarified, that “For histological and immunohistochemical staining, as well as proteomic analysis tissue sections from the exact same animals were used.” (line: 481-482)
Furthermore, we discuss the relevance of the sex in the discussion of the revised version: “In this study, porcine tissue of unknown sex was used. Several studies have investigated differences between male and female human cartilage, showing variations in cartilage thickness and joint size [30,31]. Furthermore, sex-specific differences in phenotype and progression of OA have been reported; however, additional factors such as age, BMI, and lifestyle are also highly relevant. These studies do not indicate substantial differences in cartilage composition between sexes [32–34]. Our results show consistent cartilage thickness across 33 porcine joints with low variance, suggesting that sex does not have a major impact in this context. Despite the unknown sex of the animals, significant results were obtained. Nevertheless, this study is limited by the fact that sex-specific effects cannot be excluded, which should be considered in future research.” (line: 339-348)
Comment 2: Although porcine cartilage is used as an alternative model for human tissue, the extrapolation to human therapeutic strategies should be discussed with in a bit more detailed way. The authors should specifically address differences in ECM composition and mechanical loading patterns between porcine and human knees, with references to comparative studies. The limitations relating to species differences deserve expanded discussion in the conclusion and discussion sections.
Response 2: We appreciate the reviewers comment and discussed the differences between human and porcine cartilage in a more detailed way. We have added the following to the discussion: “A review by Cone et al. (2017) summarized various porcine studies and, despite of some limitations, concludes that porcine models are gaining importance as translational models in musculoskeletal tissue engineering and regenerative medicine, and are also well-suited to address research questions in the field of biomechanics [41]. Porcine models are also suitable for modelling the human OA phenotype due to their high re-producibility [42,43]. Ronken et al. (2012) compared the biomechanical properties of healthy human and porcine cartilage. They concluded that, although differences in loss angle represent a limitation, porcine cartilage remains a suitable model for mechanical studies, including tissue engineering applications, particularly regarding dynamic modulus and dynamic stiffness [44].” (line: 353-363)
We have also emphasized the importance of considering the limitations of cross-species models in the conclusion: “Due to the similarities between porcine and human AC in ECM composition, anatomy and loading, our findings may offer insights relevant for optimizing OA treatment in humans. Nevertheless, certain limitations must be considered, especially with cross-species models, which are addressed in the discussion.” (line: 645-648)
Comment 3: The statistical approach is generally well described, but the manuscript would benefit from inclusion of more detail about the selection and validation of statistical tests, effect sizes, and corrections for multiple comparisons (e.g., clearly explain use of FDR thresholds and ANOVA randomisations). Furthermore, I was wondering why did the authors only use proteins without missing data. I’m sure with proper and widely accepted imputation methods, they could have identified more (and potentially novel or exciting) proteins in this dataset. I was also wondering whether the authors performed principal component analysis (PCA) on the samples and how did (would) the samples cluster/separate on such a plot?
Response 3: We thank the reviewer for these constructive comments and have expanded the description of our statistical analyses in the revised manuscript:
“Further analyses were performed in Perseus (v1.6.5.0) and InstantClue (v0.10.10.20211105). ANOVA analyses and unpaired Student’s t-tests were carried out with s0 = 0.1, permutation-based FDR < 0.05, and 500 randomisations. Gene Ontology term enrichment was performed on the highest-abundance proteins per group using g:Profiler (g:SCS threshold < 0.05). To assess directional trends, we additionally applied Pearson correlation. ANOVA was used to test for general differences between groups, whereas t-tests were used for specific un-paired comparisons. Data values in tables are logâ‚‚-transformed, and values shown in heatmaps are z-score normalised.” (line: 582-582)
As requested, we now clarify our measure of effect size: logâ‚‚ fold change was used for group comparisons, while Pearson correlation coefficients were used to assess trends. The parameter s0 = 0.1 was applied solely as a variance regularisation factor in the t-tests to stabilise the statistics and is not itself an effect size.
To clarify, the complete protein list, including proteins with missing values, is provided in the supplementary material. For statistical testing, however, we applied a filter of 70% completeness to ensure sufficient statistical power, given the small group size (n = 3 per group). Several robust imputation strategies exist (e.g., random forest imputation or sliding-window approaches [Schmidt et al. 2024; DOI: 10.1016/j.celrep.2024.114374]). Random forest imputation with standard parameters (downshift 2.6, width 0.02) did not improve the identification of significantly altered proteins (see PCA in Figure 1B), mainly due to the large proportion of missing values. The sliding-window method was also not suitable here, as no consistent directional trend in protein abundance was observed. More generally, imputation methods are most effective in datasets with larger group sizes, whereas with only three replicates per group, imputation introduces substantial noise rather than improving statistical power. For these reasons, we prioritised the filtered dataset to focus on robust candidates rather than risk artefacts from imputation. Nonetheless, we explored proteins with increasing or decreasing abundance despite missing values using Pearson correlation. In our dataset, however, the most relevant proteins already showed complete data coverage.
We also performed PCA both with and without imputation. PCA on the filtered data (Figure 1A) showed clearer separation, whereas PCA on imputed data (Figure 1B) resulted in poorer clustering due to the large proportion of values imputed below the detection limit. This comparison highlights our rationale for filtering rather than relying solely on imputation, as additional imputation did not improve statistical power. We note that, while PCA without imputation shows mild clustering of the samples, no clear directional trend (e.g., along a principal component axis) could be identified.
We again thank the reviewer for pointing this out and as we fully agree that this an important aspect, the PCA results have been included in the supplementary material.
Comment 4: For the western blot analysis, semi-quantification methodology (densitometry, normalization) should be performed and described in detail, rather than just visually compared.
Response 4: We agree with the reviewer on this important point. The band intensities were evaluated by normalization using sample L2 as a reference (L2 = 1). The normalized values are presented graphically, with standard deviations from n = 3 shown as error bars (Figure 5) (line: 267). Statistical analysis was performed using one-sample t-tests, and the results are provided both in the figure legend (line: 277-278) and in the main text (line: 283, 289-292, 296-297). The description of the evaluation procedure has been revised accordingly in the Methods section (line: 630-632).
Comment 5: The criteria for selecting ECM proteins for clustering and detailed analysis (Figures 3 and 4) should be more explicitly justified. How many clusters were used? Why were these particular proteins (matrilin-3, decorin, COMP, fibrillin-1) chosen, and were other relevant ECM proteins investigated but excluded due to technical or biological reasons?
Response 5: We acknowledge the reviewer’s comment and, as requested, would like to provide more background information. Our primary interest was in local differences within cartilage tissue, particularly between the two condyles (medial vs. lateral). For this reason, we specifically filtered the proteomic data to identify proteins that differed significantly between the condyles while following the same trend (Figure 3).
In contrast, Figure 4 was deliberately created with this selected set of proteins, as these were of particular interest to us because of their known function in cartilage and their role in OA. In addition to these biological reasons outlined in the manuscript, there were also technical considerations. In preliminary work, we tested both antibodies generated by ourselves and commercially available ones (e.g. Matrilin-1 and 4, collagen type IX, X, XII and XIV). However, even though most antibodies were validated in knout animals not all of the available antibodies worked reliably in porcine samples limiting the selection.
We also clarified those reasons for the readers in the discussion: “Besides COL II and PGs as the main ECM components, we focused on COMP, decorin, matrilin-3, and fibrillin-1. While the role of fibrillin-1 in cartilage and OA is not yet fully understood [55,56], matrilin-3 [57,58], decorin [59], and COMP [29,60] are known to play a role in OA progression and therefore of great interest. Other proteins such as matrilin-1 and -4, or collagens type IX, X, XII, and XIV, could not be included due to the lack of suitable antibodies, which represents a limitation of this study.” (line: 394-399)
Comment 6: The manuscript claims raw data is available in supplementary material; however, that was only an Excel export. The manuscript also states that MS data are available in the ProteomeXchange Consortium via the PRIDE partner repository. However, no accession number was given.
Response 6: Thank you very much for the helpful hint. The raw data are deposited on PRIDE partner repository with the accession number PXD068032 and will be publicly available at the date of publication. For reviewers and editors, the data can be found with the following log in credentials: Username: reviewer_pxd068032@ebi.ac.uk; Password: dNDzWXmMPDap
We provided this information in the Methods section as follows: “The mass spectrometry proteomics data have been deposited to the ProteomeXchange Consortium via the PRIDE partner repository with the dataset identifier PXD068032.” (line: 570-572) Besides providing the accession number, we also included the filtered data with student t-test and ANOVA to the supplement.
Please see attachment

Reviewer 2 Report
Comments and Suggestions for Authors
This study takes a close look at differences within porcine femoral cartilage by combining histology, immunohistochemistry, Western blotting, and proteomics. The authors show that the medial side of the cartilage is thicker and contains higher levels of important extracellular matrix proteins. By bringing together structural, biochemical, and proteomic evidence, the work highlights meaningful site-specific differences in cartilage biology that could inform approaches in tissue engineering.
While the authors have done a great job at demonstrating the differences in porcine femoral cartilage based on region, relating these differences to mechanical loading purely based on the literature and proteomics studies might not be enough for this publication.
Can the authors add some validation or functional studies to demonstrate how these regional changes are associated with mechanical load?
Author Response
Comment 1: This study takes a close look at differences within porcine femoral cartilage by combining histology, immunohistochemistry, Western blotting, and proteomics. The authors show that the medial side of the cartilage is thicker and contains higher levels of important extracellular matrix proteins. By bringing together structural, biochemical, and proteomic evidence, the work highlights meaningful site-specific differences in cartilage biology that could inform approaches in tissue engineering.
While the authors have done a great job at demonstrating the differences in porcine femoral cartilage based on region, relating these differences to mechanical loading purely based on the literature and proteomics studies might not be enough for this publication. Can the authors add some validation or functional studies to demonstrate how these regional changes are associated with mechanical load?
Response: We thank the reviewer for the positive feedback and for raising this concern. From our perspective, there are sufficient published studies which we can refer to without performing own loading experiments. We acknowledge that this aspect may not have been discussed in sufficient depth in the original manuscript and apologize for this. Accordingly, in the revised version we have included additional studies on this topic and discussed them in greater detail:
“Ronken et al. (2012) compared the biomechanical properties of healthy human and porcine cartilage. They concluded that, although differences in loss angle represent a limitation, porcine cartilage remains a suitable model for mechanical studies, including tissue engineering applications, particularly regarding dynamic modulus and dynamic stiffness [44].” (line: 359-363)
“Two studies have shown that higher forces are typically transmitted through the medial compartment of the human knee during gait [48,49]. Shiomi et al. (2010) analysed porcine knee joints using T2 (transverse relaxation time) mapping and demonstrated that higher T2 values occur in the lateral condyle under loading. In contrast, the medial condyle, which is generally subjected to higher loads, exhibits a denser collagen net-work, resulting in lower T2 values [50].” (line: 371-377)
“Overall, these results demonstrate suggest that higher quantities of several ECM proteins in the medial joint compartment directly may contributes to its enhanced ability to withstand locally increased mechanical loads, thereby maintaining cartilage stability and integrity. The present analyses align with Shiomi et al. (2010), who reported that differences between the medial and lateral compartments occur in porcine knee joints, and are consistent with the assumption that medial cartilage may be more densely structured to withstand higher mechanical loads [50].” (line: 430-436)
Please see attachment

Reviewer 3 Report
Comments and Suggestions for Authors
This study provides a comprehensive analysis of porcine articular cartilage (AC) from eight defined anatomical sites in the knee, focusing on cartilage thickness, proteoglycan (PG) content, and extracellular matrix (ECM) protein distribution. By integrating histological evaluation with proteomic profiling and validating results through immunohistochemistry and Western blotting, the authors demonstrate significant regional variations between medial and lateral compartments, as well as zone-specific localization patterns of key ECM proteins. These findings are of considerable importance for understanding how mechanical loading shapes cartilage composition and for informing future tissue engineering and cartilage repair strategies.
While the manuscript is well-designed and presents novel and relevant findings, several issues should be addressed before it can be considered for publication:
- In the introduction, the following references on bone/cartilage regeneration might be helpful to provide more information and benefit the future readers: e.g., The high-strength and toughness Janus bionic periosteum matching bone development and growth in children. Composites Part B. 2023, 256: 110642./ Asian facial recontouring surgery. Plast Aesthet Res. 2023;10:59./ Up IGF-I via high-toughness adaptive hydrogels for remodeling growth plate of children. Regenerative Biomaterials, 2025, 12, rbaf004. / Alveolar cleft reconstruction with vomerine bone: two surgical procedures in one step: a case series. Plast Aesthet Res. 2023;10:16./ Electrospun fiber-based immune engineering in regenerative medicine, Smart Medicine 2024, 3 (1), e20230034./ Bone regeneration and antibacterial properties of calcium-phosphorus coatings induced by gentamicin-loaded polydopamine on magnesium alloys. Biomedical Technology 2024, 5, 87-101.
- In Figure 1c/d, the differences between medial and lateral compartments were analyzed using a t-test (reported as p = 4.46E-21, with p < 0.05 as the significance threshold). Considering the hierarchical structure of multiple animals and multiple positions (L1–L4 / M1–M4), as well as the correlation within each animal, the use of a simple t-test is not appropriate.
- The manuscript relates the higher ECM protein abundance in the medial compartment to adaptation to mechanical loading; however, no biomechanical or functional measurements were included to support this interpretation.
- In Figure 1, please clearly indicate the units (µm or mm) for the data. The figure legend states “thickness n=30 (Methods),” but the meaning of n is unclear. Please specify in the Methods and figure legend whether n refers to the number of animals, slices per animal, or measurements per anatomical point, and how the total of 30 was obtained.
- In Figure 5, please provide biological replicates and quantitative analysis of band intensity (mean ± SD, with n and statistical test indicated).
- In Figure 6, it is recommended to include nuclear staining (e.g., DAPI) to identify chondrocytes, or state clearly in the figure legend why nuclear staining was not used. In addition, each micrograph should contain a visible scale bar and magnification.
- Figure 7 is critical for understanding the localization of L1–L4 / M1–M4. It is recommended to move this figure closer to Figure 1 (e.g., as Figure 1b), and to annotate weight-bearing versus non-weight-bearing regions in the schematic if applicable.
- In Table 1, several antibodies are listed with a “h-” (human) prefix, whereas the samples are porcine. Please clarify the cross-reactivity validation (manufacturer’s datasheet, literature RRID, or experimental evidence), and provide catalog numbers, RRIDs, and batch numbers where possible.
Author Response
This study provides a comprehensive analysis of porcine articular cartilage (AC) from eight defined anatomical sites in the knee, focusing on cartilage thickness, proteoglycan (PG) content, and extracellular matrix (ECM) protein distribution. By integrating histological evaluation with proteomic profiling and validating results through immunohistochemistry and Western blotting, the authors demonstrate significant regional variations between medial and lateral compartments, as well as zone-specific localization patterns of key ECM proteins. These findings are of considerable importance for understanding how mechanical loading shapes cartilage composition and for informing future tissue engineering and cartilage repair strategies.
While the manuscript is well-designed and presents novel and relevant findings, several issues should be addressed before it can be considered for publication:
- Point-by-point response to Comments and Suggestions for Authors
Comments 1: In the introduction, the following references on bone/cartilage regeneration might be helpful to provide more information and benefit the future readers: e.g., The high-strength and toughness Janus bionic periosteum matching bone development and growth in children. Composites Part B. 2023, 256: 110642./ Asian facial recontouring surgery. Plast Aesthet Res. 2023;10:59./ Up IGF-I via high-toughness adaptive hydrogels for remodeling growth plate of children. Regenerative Biomaterials, 2025, 12, rbaf004. / Alveolar cleft reconstruction with vomerine bone: two surgical procedures in one step: a case series. Plast Aesthet Res. 2023;10:16./ Electrospun fiber-based immune engineering in regenerative medicine, Smart Medicine 2024, 3 (1), e20230034./ Bone regeneration and antibacterial properties of calcium-phosphorus coatings induced by gentamicin-loaded polydopamine on magnesium alloys. Biomedical Technology 2024, 5, 87-101.
Response 1: We would like to thank the reviewer for the valuable suggestions. We have incorporated two relevant references in the Introduction to provide the reader with additional information on cartilage and bone regeneration: Up IGF-I via high-toughness adaptive hydrogels for remodeling growth plate of children. Regenerative Biomaterials, 2025, 12, rbaf004 and Electrospun fiber-based immune engineering in regenerative medicine, Smart Medicine 2024, 3 (1), e20230034
“Tissue engineering and regenerative medicine are becoming increasingly important, both in the context of cartilage/ bone regeneration and in the broader field of biomedical research [23,24].” (line: 92-94)
Comment 2: In Figure 1c/d, the differences between medial and lateral compartments were analyzed using a t-test (reported as p = 4.46E-21, with p < 0.05 as the significance threshold). Considering the hierarchical structure of multiple animals and multiple positions (L1–L4 / M1–M4), as well as the correlation within each animal, the use of a simple t-test is not appropriate.
Response 2: We thank the reviewer for pointing out that issue. To address this point, we consulted a colleague with specific experience in statistical analysis. In the revised manuscript, we applied a linear mixed-effects model instead of the previously used t-test. In this model, the individual animals were included as a random intercept, while side (medial vs. lateral) and position (locations 1–4) were treated as fixed effects. The model showed that medial cartilage was on average 0.68 mm thicker than lateral cartilage (β = –0.678, SE = 0.068, t(149) = –9.93, p < 0.001, 95% CI [–0.813, –0.543]). Furthermore, the analysis indicated no significant differences between the individual positions (β = –0.0064, p = 0.667). The updated statistical evaluation therefore revealed that the significant difference lies between the two condyles, independent of the positions within each side.
We added a description of this analysis in the Methods section (line: 544-549) and described the results in the figure legend (line: 120-121) as well as the main text (line: 140-142 and 151-154).
Comment 3: The manuscript relates the higher ECM protein abundance in the medial compartment to adaptation to mechanical loading; however, no biomechanical or functional measurements were included to support this interpretation.
Response 3: We appreciate the reviewer’s comment and, to further support our hypothesis regarding mechanical loading, we have incorporated additional studies in the revised manuscript and discussed them in greater detail: “Ronken et al. (2012) compared the biomechanical properties of healthy human and porcine cartilage. They concluded that, although differences in loss angle represent a limitation, porcine cartilage remains a suitable model for mechanical studies, including tissue engineering applications, particularly regarding dynamic modulus and dynamic stiffness [44].” (line: 359-363)
We have also included two studies that clearly demonstrate that the medial condyle bears higher loads compared to the lateral condyle: “Two studies have shown that higher forces are typically transmitted through the me-dial compartment of the human knee during gait [48,49]. Shiomi et al. (2010) analysed porcine knee joints using T2 (transverse relaxation time) mapping and demonstrated that higher T2 values occur in the lateral condyle under loading. In contrast, the medial condyle, which is generally subjected to higher loads, exhibits a denser collagen net-work, resulting in lower T2 values [50].” (line: 371-377)
“The present analyses align with Shiomi et al. (2010), who reported that differences between the medial and lateral compartments occur in porcine knee joints, and are consistent with the assumption that medial cartilage may be more densely structured to withstand higher mechanical loads [50].” (line: 433-436)
Comment 4: In Figure 1, please clearly indicate the units (µm or mm) for the data. The figure legend states “thickness n=30 (Methods),” but the meaning of n is unclear. Please specify in the Methods and figure legend whether n refers to the number of animals, slices per animal, or measurements per anatomical point, and how the total of 30 was obtained.
Response 4: We agree with the reviewer that this aspect could be improved. We have now added units in Figure 1 wherever necessary. In addition, we clarified in both the figure legend and the Methods section that n = 33 refers to the number of individual knee joints and specified how many measurements were performed per joint. Finally, we corrected the number from 30 to 33.
In a previous study, a significant difference in cartilage thickness between lateral and medial compartment has been shown already when comparing only 4 samples (Ma et al. 2024, DOI: 10.1007/s10561-024-10126-3). Sample size estimation was performed using a one-tailed paired t-test based on the published data of Ma et al. (2024). Assuming a correlation of 0.3 between the two groups, with α = 0.05 and β = 0.1 (power = 90 %), the required sample size was N = 20. In our case, the samples were by-products of food production and were available in sufficient numbers. Additional knee joints were also accessible for other experiments, which allowed us to include them in the thickness measurements as well.
In the Results section, we clarified how the total number of samples (n = 33) was derived: “Based on the study by Ma et al. (2024), we conducted a power analysis (one-tailed paired t-test; correlation = 0.3; α = 0.05; β = 0.1), which indicated a required sample size of n = 20 [27]. As sufficient specimens were available, the sample size was increased to 33, providing a calculated power of 100 %. (Figure 1).” (line: 109-112)
Comment 5: In Figure 5, please provide biological replicates and quantitative analysis of band intensity (mean ± SD, with n and statistical test indicated).
Response 5: We appreciate that this point was raised and we agree with the reviewer. The replicates have been specified in the figure legend: n = 3 refers to samples from three individual knee joints (line: 269). Band intensities were evaluated by normalization using sample L2 as a reference (L2 = 1). The normalized values are presented graphically, with standard deviations from n = 3 shown as error bars (line: 267). Statistical analysis was performed using one-sample t-tests, and the results are provided both in the figure legend (line: 277-278) and in the main text (line: 283, 289-292, 296-297). The description of the evaluation procedure has been revised accordingly in the Methods section (line: 630-632).
Comment 6: In Figure 6, it is recommended to include nuclear staining (e.g., DAPI) to identify chondrocytes, or state clearly in the figure legend why nuclear staining was not used. In addition, each micrograph should contain a visible scale bar and magnification.
Response 6: We understand the reviewer’s point and would like to provide further clarification. In this figure, we deliberately chose not to include nuclear staining. This decision was based on technical considerations, as staining of paraffin-embedded cartilage samples results in increased autofluorescence within the wavelength range of DAPI and related dyes that might distract the reader from the specific matrix staining We have now clarified this in the figure legend: “Nuclear staining was not included due to autofluorescence in paraffin-embedded tissue sections” (line: 204-305). In addition, scale bars have been inserted into each microscopic image, and the magnification has been specified in the figure legend. (line: 305)
Comment 7: Figure 7 is critical for understanding the localization of L1–L4 / M1–M4. It is recommended to move this figure closer to Figure 1 (e.g., as Figure 1b), and to annotate weight-bearing versus non-weight-bearing regions in the schematic if applicable.
Response 7: We thank the reviewer for this valuable suggestion. Figure 7 has been removed from the Methods section and integrated into Figure 1 for better understanding the localisation of L1–L4 / M1–M4 (line: 112). We decided against annotating weight-bearing versus non-weight-bearing regions in the schematic as no load-bearing experiments were performed in this study and previously published data were used instead. Furthermore, load distributions in the knee joint have mainly been described in human studies, and given the cross-species nature of the data, we aimed to avoid misleading the reader by combining them into a single figure.
To nevertheless address this point appropriately, we described the load distribution across the positions in the discussion and referred to the relevant studies: “While cartilage thickness did not differ significantly between positions 1 – 4, previous studies indicate regional variations in load distribution in human AC [45,46]. Accordingly, positions M2/L2 and M3/L3 in our model are likely subject to higher loads.” (line: 378-380)
Comment 8: In Table 1, several antibodies are listed with a “h-” (human) prefix, whereas the samples are porcine. Please clarify the cross-reactivity validation (manufacturer’s datasheet, literature RRID, or experimental evidence), and provide catalog numbers, RRIDs, and batch numbers where possible.
Response 8: We appreciate this valuable point and have addressed it in the revised manuscript. The following section has been added to the Methods: “The listed antibodies are based on human protein sequences, which does not preclude their applicability to proteins of other species. According to the datasheet, the collagen type II antibody detects not only human protein but also rat, bovine, and rabbit. To ensure that all antibodies were suitable for porcine samples, the protein sequences were retrieved from the https://www.uniprot.org/ database and aligned using the NIH BLASTp (https://blast.ncbi.nlm.nih.gov/Blast.cgi) to determine their homology.” (line: 523-528)
In the Results section, we also reported the outcomes of the sequence alignment: “The primary antibodies used for staining were developed based on human protein sequences. To ensure that they also detect porcine proteins, the human and porcine sequences were compared. All analysed proteins exhibited a very high proportion of identical amino acid sequences, making it highly likely that the antibodies also bind to the corresponding porcine proteins: Decorin = 91.1 %; Matrilin-3 = 88.2 %; Fibrillin-1 = 96.9 %; Collagen type II = 97.7 % and COMP = 93.3 % (supplement 2).” (line: 310-315)
Please see attachment

Round 2
Reviewer 1 Report
Comments and Suggestions for Authors
Thank you for revising the manuscript carefully according to my comments and suggestions. I appreciate your effort and the improvements made to the work.